# Anomalous Self-Organization in Active Piles

**DOI:** 10.3390/e25060861

**Published:** 2023-05-27

**Authors:** Morteza Nattagh-Najafi, Mohammad Nabil, Rafsun Hossain Mridha, Seyed Amin Nabavizadeh

**Affiliations:** Department of Mechanical Engineering, University of Akron, Akron, OH 44325, USArm283@uakron.edu (R.H.M.)

**Keywords:** self-organized active pile, stretched exponential distribution, Levy-stable distribution, 05., 05.20.-y, 05.10.Ln, 05.45.Df

## Abstract

Inspired by recent observations on active self-organized critical (SOC) systems, we designed an active pile (or ant pile) model with two ingredients: beyond-threshold toppling and under-threshold active motions. By including the latter component, we were able to replace the typical power-law distribution for geometric observables with a stretched exponential fat-tailed distribution, where the exponent and decay rate are dependent on the activity’s strength (ζ). This observation helped us to uncover a hidden connection between active SOC systems and α-stable Levy systems. We demonstrate that one can partially sweep α-stable Levy distributions by changing ζ. The system undergoes a crossover towards Bak–Tang–Weisenfeld (BTW) sandpiles with a power-law behavior (SOC fixed point) below a crossover point $\zeta <\zeta^{*}\approx 0.1$.

## 1. Introduction

Self-organized criticality is a widespread phenomenon referring to how individual constituents organize their interactions to create critical behaviors. This concept can be identified in various natural phenomena, such as neural dynamics in the brain [1,2,3,4,5,6], rough surfaces and the roughening process [7], liquid foams [8], sheared suspensions [9], atmospheric cascades [10,11], earthquakes [12,13,14,15], stock markets [16,17,18,19,20], social sciences [21,22,23,24], astrophysics [25], knowledge creation [26], rainfall [27], electronic avalanches in electron gas [28], vortices in superconductors [29,30], real sandpiles and rice piles [31], and self-organized Levy flights [32]. To be more precise, the models describing earthquakes [33], forest fires [34], and neuronal avalanches [35,36,37] are not strictly characterized as self-organized criticality (SOC). Instead, they exhibit a form of approximate SOC, often referred to as self-organized quasicriticality (SOqC) or homeostatic criticality (see also [37,38,39]). Active matter systems, such as self-propelled colloids and bacterial systems [40,41,42,43], can show self-organized criticality. Examples of self-organized criticality in active matter systems include the universal scaling laws for activity bursts of collective cell migration [44], where intracellular biological mechanisms play a dominant role, and the scale-free avalanches in active systems modeled by run-and-tumble disks [45]. Despite its observation in numerous active matter systems, the mechanism behind criticality in these systems remains an open question.

The interaction among the active constituents of the system leads to critical states in active Brownian motion [43,46,47] and active lattice gas [42,43], demonstrating critical behavior in specific points or within limited segments of their phase space. Additionally, in an active matter system mapped to an elastic interface moving in the background of obstacles, criticality is attributed to the depinning transition [45]. The active Ising model [48], the modified flocking transition model [49], swarming transition in a modified Vicsek model [50,51], and vapor–liquid transition in active Lennard-Jones fluid [52] are some examples of attempts at understanding criticality in active matter systems.

This letter introduces an asymptotically self-organized critical active model inspired by a pile of ants as active particles. In our model, a pile of active agents (ants) wander around on top of each other within the system. If the population of ants in a region exceeds a threshold, *toppling* takes place (similar to sandpile models [53]), and the local column of agents overflow to the neighboring regions or leave the system. This active pile shows an anomalous self-organized phase and provides a connection to α-stable Levy systems.

## 2. Methods and Results

We define an active pile model on a square lattice of size L×L, and ${\left\{n_{i}\right\}}_{i=1}^{L^{2}}\in \left\{1,2,3,4\right\}$ represents the number (*height* of the column) of ants at each lattice site. We performed simulations for different lattice sizes, defined as L=32,64,128, and 256. When the height at a given site *i* exceeds the threshold of four, it is called *unstable*, and toppling takes place: the site loses four ants, which are either distributed to neighboring sites or wasted at the lattice boundaries. More precisely, during the toppling of site *i*, $n_{j}\to n_{j}+\Delta_{i,j}$, where $\Delta_{i,j}=-4$ if i=j, +1 if *i* and *j* are neighbors, and 0 otherwise ($\forall j\in [1,L^{2}]$). To initiate the system, we start from a random pile configuration and periodically add ants to the active pile at a constant rate. Between two successive additions, a fraction of ants travel to their neighboring site (they cannot leave the lattice boundaries) such that a total of *N* active movements happen. The *strength* of the pile’s activity is quantified by $\zeta \equiv N/L^{2}$ and is in the range $\left[10^{-4},20\right]$. The asymptotic limit ζ→0 corresponds to an ordinary BTW sandpile model [54] (BTW here refers to the inventors of this model, i.e., Bak, Tang, and Weisenfeld), while ζ→∞ corresponds to a hyper-active pile. The reason why we call the latter case “hyper-active” is because of the prominence of self-propelled activity with respect to toppling activity in this limit. To understand this, we note that since toppling is prohibited during active motions (once $N=\zeta L^{2}$ active motions are completed, the unstable sites are allowed to topple), the typical duration of active motions is much larger than the typical duration of the toppling process in this limit. Let us define $n_{\mathrm{top}}$ as the number of toppling events in an avalanche and ξ as $n_{\mathrm{top}}/N$. Typically, $n_{\mathrm{top}}$ is proportional to $L^{2}$ per avalanche, leading to ξ∝1/ζ, so that as ζ approaches infinity, ξ approaches zero. This tells us that in this limit, the activity dominates the toppling dynamics. Furthermore, defining $n_{l}$ as the typical number of ants leaving the system in an avalanche and assuming that $n_{l}\propto n_{\mathrm{top}}$, $n_{l}/N$ approaches zero in the limit of large ζ. This shows that the system corresponds to a conservative active pile with a small leakage factor ξ. Thus, our model is expected to fit better with a hyper-active model, in which particles only undergo active motions with minimal leakage proportional to ξ, which we call a hyper-active pile.

During an “active movement”, an ant is randomly selected to move to a neighboring site within the lattice. This can cause the site to become unstable and topple. The strength of the activity can result in multiple toppling events throughout the system between successive additions of ants. The series of toppling events that occur between two successive ant injections is referred to as an *avalanche*. The focus of the study is on two measurements of the avalanche: “avalanche mass” (m), which is the total number of toppled sites, and “avalanche size” (s), which is the total number of toppling events that occur during the avalanche. We also focus on geometrically observable properties that will be defined later.

The average height, $\bar{n}:=L^{-2}\sum_{i=1}^{L^{2}}n_{i}$, is a useful metric for understanding the dynamic properties of the active pile. It is defined as the sum of heights of all sites divided by the total number of sites. The number of ants in the system is proportional to time *T*, with a proportionality constant set to one. Figure 1 displays the time evolution of $\bar{n}$. Initially, the system explores various transient configurations, causing $\bar{n}$ to increase over time until it reaches dynamic transition point $T_{\zeta }^{*}$. Beyond this point, the system reaches a stationary regime consisting of recurrent configurations. Our statistical tests indicate that $\bar{n}$ remains nearly constant in the stationary regime, regardless of the initial configuration for given ζ and *L*. The ant pile dynamically self-organizes into the same recurrent states, indicating that this recurrent state is a fixed point of the ant pile dynamics. We refer to this regime as the *anomalous self-organized* state, where in analogy with glassy dynamics, the term “anomalous” reflects the presence of stretched exponential distributions (SEDs), which will be further discussed.

In the stationary regime, $\bar{n}\left(T\right)≃{\langle \bar{n}\rangle }_{T}$, where ${\langle \cdot \rangle }_{T}$ is the time average in the stationary regime. Figure 1 displays the time dependence of $\bar{n}$, with the insets showing $T_{\zeta }^{*}$ and ${\langle \bar{n}\rangle }_{T}$ in terms of ζ and *L*. Notably, since $T_{\zeta =0}^{*}\propto L^{2}$ [53] (corresponding to the standard BTW model), we rescaled the values in the plots as $T_{\zeta }^{*}/L^{2}$ for ease of comparison and to keep the numbers reasonable. We also performed this to enable the values of ${\langle \bar{n}\rangle }_{T}$ to be plotted on the same graph. Regarding ζ (right inset in Figure 1), $T_{\zeta }^{*}/L^{2}$ exhibits fluctuations with no significant trend, while ${\langle \bar{n}\rangle }_{T}$ decreases with ζ and saturates to a final value for large ζ. An exponential function is defined to fit the tail of ${\langle \bar{n}\rangle }_{T}$ in the limit of large ζ.
(1)$$f\left(\zeta \right)=-A\exp\left[-\gamma_{f}\zeta \right]+f_{\infty }$$
where A>0 is a constant, $f_{\infty }\equiv \lim_{\zeta \to \infty }f\left(\zeta \right)$ represents the asymptotic value of *f* in the hyper-active pile limit, and $\gamma_{f}$ is the corresponding exponent. Two important limits are $\lim_{\zeta \to 0}{\langle \bar{n}\rangle }_{T}\approx 3.2$, which corresponds to the BTW model limit [53], and ${\langle \bar{n}\rangle }_{T}\approx 2.12$ for the ζ→∞ limit. The left inset in Figure 1 presents these functions plotted against 1/L. This representation aids in extrapolating the trend of the functions as they approach the origin. Interestingly, both ${\langle \bar{n}\rangle }_{T}$ and $T_{\zeta }^{*}/L^{2}$ exhibit only small dependence on *L*.

Active motions destroy the criticality of a system, leading to distinct ζ-dependent characteristics of the avalanche mass and size. This phenomenon breaks the scale-invariance property that is typical of (SOC) systems [53,55]. Unlike the power-law behavior observed in critical systems [56], the distribution function of these observables in the active system is described by a Gaussian distribution, as shown in Figure 2a. Specifically, for x=m (avalanche mass) or *s* (avalanche size), we can write
(2)$$p\left(x,\zeta \right)\propto \exp\left[-\frac{{\left(x-\mu_{x}\left(\zeta \right)\right)}^{2}}{2\sigma_{x}{\left(\zeta \right)}^{2}}\right],$$
where $\mu_{x}\left(\zeta \right)$ is the characteristic mean value of *x* in the anomalous self-organized state and $\sigma_{x}{\left(\zeta \right)}^{2}$ is its variance. The research findings suggest that active motions decrease the occurrence of both small and large (rare) avalanches. Specifically, the frequency of small events decreases because active motions drive the sites towards stability, while large events are avoided by releasing highly active sites. $\mu_{x}$ shows a non-monotonic behavior in terms of ζ, i.e., it grows for a small ζ regime and decreases for larger ζ. Meanwhile $\sigma_{x}$ goes to infinity in a power-law fashion $\zeta^{-\chi }$ as ζ→0, where χ=0.23±0.04 is an exponent (inset in Figure 2b). Both quantities exhibit power-law behavior regarding system size. These observations indicate that as ζ approaches zero, the fluctuations of the system become unbounded, and critical behaviors emerge. This leads to the system reaching a self-organized critical (SOC) state that corresponds to the BTW sandpile fixed point [53].

The critical properties of a pile are modulated by active motions acting as an under-threshold toppling factor (known as under-threshold spiking in neuronal systems [57]), as demonstrated by our results, including Equation (Equation 2). This modulation is beyond a simple cut-off in the range of the critical behavior observed in dissipative sandpiles [56]. We observed that the change in geometrical observables is particularly intriguing; stretched exponential distributions (SEDs) emerge and exhibit a transition to a power-law regime as the limit ζ→0 is approached. To demonstrate this, the global properties of sub-avalanches (connected components of avalanches) are examined, including y=sm (sub-avalanche mass), *l* (loop length of sub-avalanche boundaries), and $r_{g}^{2}$ (the square gyration radius of sub avalanche boundaries). To define these quantities, let us define $G=\left\{i|i\in \text{sub-avalanche}\right\}$ and $\partial_{\mathrm{outer}}G$ as its outer boundary (which separates the cluster from infinity). Then, $sm=\sum_{i}\delta_{i,G}$, where $\delta_{i,G}=1$ if *i* belongs to *G* and zero otherwise, and $l=\sum_{i}\delta_{i,\partial_{\mathrm{outer}}G}$. The gyration radius is then defined as $r_{g}^{2}=\frac{1}{l}\sum_{i\in \partial_{\mathrm{outer}}G}{\left|{\vec{r}}_{i}-{\vec{r}}_{\mathrm{com}}\right|}^{2}$, where ${\vec{r}}_{i}$ is the position vector of site *i*, ${\vec{r}}_{\mathrm{com}}=\frac{1}{l}\sum_{i\in \partial_{\mathrm{outer}}G}{\vec{r}}_{i}$ is the loop center of mass, and $\left|\ldots \right|$ shows the vector length. It should be noted that the presence of active motions results in the avalanches being fragmented and comprising multiple disconnected sub-avalanches. We depict the distribution function for these quantities (p(y,ζ)) in Figure 3a, where we show $\ln\left(-\ln\frac{p(y,\zeta )}{p_{0y}}\right)$ in terms of lny ($p_{0y}$ is a proportionality constant), which is linear. This confirms that they obey the standard SED relationship:(3)$$p\left(y,\zeta \right)=p_{0y}\exp\left[-{\left(\lambda_{y}\left(\zeta \right)y\right)}^{\alpha_{y}\left(\zeta \right)}\right],$$
where $\alpha_{y}\left(\zeta \right)$ is the ζ-dependent *stretching exponent*, which gives the function a slower decay than a pure exponential function (i.e., when $\alpha_{y}=1$), and $\lambda_{y}\left(\zeta \right)$ is an SED decay factor. The linear fittings are valid down to a *crossover region*, i.e., $\zeta >\zeta^{*}\approx 0.1$. Below this point, the system experiences a crossover to BTW-like behavior, which will be explained later. For example, at ζ=0.1, a change in slope is observed, as shown in Figure 3a, which is characteristic of crossover phenomena [56].

SEDs are well-known for slow relaxation in complex condensed systems [58], as demonstrated in many physical systems. The Fourier transformation of SEDs was first applied by Williams et al. [59] to describe the dielectric spectra of polymers, resulting in the Kohlrausch–Williams–Watts function. It is observed in materials science to describe the relaxation behavior of glasses and polymers, in organic liquids, in neuroscience to model the decay of synaptic transmission, and in finance to describe the distribution of stock price fluctuations. SEDs have also been observed in turbulence and earthquakes. Therefore, the Fourier transformation of SEDs is a widely used tool to describe various phenomena across multiple disciplines [60,61,62,63,64,65,66,67]. SEDs have been the focus of researchers and remain one of the oldest unresolved problems in the field (see [68] for a good review). The stretching exponent is a crucial parameter for characterizing a wide range of physical systems and provides insights into temporal dynamics and complexity. When the stretching exponent is different from one, it indicates that different parts of the system decay at different rates, thereby quantifying the degree of heterogeneity or complexity within the system. Various factors, such as spatial variations or multiple timescales, can lead to this heterogeneity or complexity within the system [69]. SEDs are related to various distributions in the context of stochastic processes, such as Fox–Wright psi distributions, ${}_{p}\Psi_{q}$ (which are themselves related to the anomalous diffusion problem [70]); fractional Brownian motion [71]; and asymmetric α-stable Levy distributions, where these distributions can quantify the degree of temporal correlations [72].

The stretching exponent quantifies the level of temporal correlation within a system. A stretching exponent smaller than one indicates a high degree of correlation, while a stretching exponent greater than one implies a low level of correlation. The behavior of α and λ for $y=sm,l,r_{g}^{2}$ in terms of ζ is shown in Figure 3b,c, and the fits are according to Equation (Equation 1). These results show that $\alpha_{y}$ is an increasing function of ζ. This indicates that as ζ increases, the decay attenuation factor of avalanches also increases. This implies that avalanches become more dissipative in this case. To comprehend this, we can think of active motion as the removal of a particle from one site and its simultaneous creation in one of its neighboring sites. The annihilation (creation) of particles in sandpiles is equivalent to considering mass (imaginary mass) in ghost field theory [56]. This reduces the correlation length and the range of the avalanches. Therefore, the larger the amount of ζ is, the lower the range of the avalanches is (see also the Laplace analysis below). The fitting results indicate that as ζ approaches infinity, $\alpha_{\mathrm{sm}}$, $\alpha_{l}$, and $\alpha_{r_{g}^{2}}$ converge to the hyper-active pile limits of 0.85±0.02, 1.04±0.02, and 0.77±0.02 with exponents $\gamma_{y}=0.074\pm 0.006$, 0.071±0.005, and 0.16±0.02, respectively. We also observed that when ζ decreases (Figure 3c), an abrupt change in behavior takes place for $\zeta <\zeta^{*}\approx 0.1$, in which α falls rapidly so that α→0 in the limit ζ→0 for all geometrical quantities. As a result, one retrieves the power-law (BTW-like) behavior for $\zeta <\zeta^{*}$, given the simple identity for small $\alpha_{y}$,
(4)$$p_{\mathrm{BTW}}\left(y\right)\equiv \lim_{\zeta \to 0}p\left(y,\zeta \right)\propto y^{-\tau_{y}}$$
where
(5)$$\tau_{y}\equiv \lim_{\zeta \to 0}\alpha_{y}\left(\zeta \right)\lambda_{y}{\left(\zeta \right)}^{\alpha_{y}\left(\zeta \right)}.$$
is the exponent in the BTW limit. This is consistent with our assertion about retrieving the SOC (BTW) fixed point in the limit ζ→0. The phase diagram of the active pile model is schematically sketched in Figure 4. In the limit ζ→0, it behaves similar to the ordinary BTW model. It undergoes a crossover to an SED regime beyond a crossover point $\zeta^{*}\approx 0.1$. For $\zeta >\zeta^{*}$, it shows an SED, and the extreme limit is observed in the limit ζ→∞.

Although Equation (Equation 3) and the previous analysis demonstrate the relationship between the stretching exponent and the attenuation of avalanches, it is beneficial to express the stretching exponent in terms of *exponentially decaying* components with well-defined attenuation factors. The Laplace transformation provides an effective way of achieving this representation of SEDs, as shown below.
(6)$$p\left(y,\zeta \right)=\int_{0}^{\infty }{\tilde{p}}_{\alpha }^{\left(y\right)}\left(s,\zeta \right)\exp\left[-s\lambda_{y}\left(\zeta \right)y\right]ds$$
where ${\tilde{p}}_{\alpha }^{\left(y\right)}\left(s,\zeta \right)$ is the “strength” of a component (Laplace component) that decays exponentially with the attenuation factor ${\left[s\lambda_{y}\left(\zeta \right)\right]}^{-1}$. The Laplace component is given using the inverse Laplace transformation, which is [58]
(7)$$\begin{array}{cc} {\tilde{p}}_{\alpha }^{\left(y\right)}\left(s,\zeta \right) & =\frac{1}{2\pi }\int_{-\infty }^{\infty }e^{-{\left(iu\right)}^{\alpha_{y}\left(\zeta \right)}}e^{isu}du\\ \; & =\frac{1}{\pi }\sum_{n=0}^{\infty }\frac{{\left(-1\right)}^{n+1}\Gamma \left(n\alpha_{y}\left(\zeta \right)+1\right)}{n!s^{n\alpha_{y}\left(\zeta \right)+1}}\sin\left(n\pi \alpha_{y}\left(\zeta \right)\right), \end{array}$$
where Γ[x] is the Gamma function [73]. It is notable that the tails of ${\tilde{p}}_{\alpha }^{\left(y\right)}\left(s,\zeta \right)$ follow the asymptotic decay behavior of $s^{-1-\alpha_{y}\left(\zeta \right)}$. In some limits, the analytical form of this function is known. For example, as indicated in Figure 3a, in the crossover region (around ζ=0.1), $\alpha_{\mathrm{sm}}$, $\alpha_{l}$, and $\alpha_{r_{g}^{2}}$ are approximately $\frac{1}{2}$, for which the analytic form of $\tilde{p}$ is known to be [58]
(8)$${\tilde{p}}_{\frac{1}{2}}^{\left\{\left(\mathrm{sm}\right),\left(l\right),\left(r_{g}^{2}\right)\right\}}\left(s,\zeta =0.1\right)\approx \frac{1}{\sqrt{4\pi s^{3}}}\exp\left[-{\left(4s\right)}^{-1}\right],$$
while $\alpha_{r_{g}^{2}}\approx 1$ is consistent with a simple exponential decay. In Figure 5, we show ${\tilde{p}}_{\alpha }\left(s,\zeta \right)$ for y=l. We observe a global decay of many independently decaying pieces (Laplace components), with *s* showing a specified decay rate. $s_{\min}$ (the minimum *s* under which ${\tilde{p}}_{\alpha }\left(s,\zeta \right)$ is zero) and $s_{\max}$ (the peak point) are pushed to the right when ζ increases, i.e., the larger the amount of ζ is, the faster the sub-avalanches decay. This aligns with our observations regarding total avalanches. As the system experiences more active motion, the avalanches become more dissipative. Furthermore, note the fat tail of ${\tilde{p}}_{\alpha }\left(s,\zeta \right)$ for $s>s_{\max}$, and note that the decay is faster for larger ζ values. A transition is seen for $s_{\max}$ in the crossover region (around ζ≈0.1, highlighted in blue in the inset in Figure 5), above which it scales with ζ as a power law.

Given that an SED is a bridge between normal and α-stable Levy distributions [72,74,75], our analysis uncovers a hidden relation between active particles and α-stable Levy distributions. Using the correspondence of the Laplace transformation and the Fourier transformation of $\exp\left(-y^{\alpha }\right)$, one finds the relation of the distributions that we found with the asymmetric Levy-stable distributions as follows [76]:(9)$${\tilde{p}}_{\alpha }^{\left(y\right)}\left(s,\zeta \right)=L\left(s;\alpha_{y}\left(\zeta \right),-\alpha_{y}\left(\zeta \right)\right),$$
where L is a generalized α-stable Levy distribution function, given by the relation
(10)$$L\left(s;\alpha ,\gamma \right)=\frac{1}{\pi }\int_{0}^{\infty }e^{-isu}e^{-u^{\alpha }e^{i\pi \gamma /2}}du.$$The fat-tailed probability distribution of the α-stable Levy process is indicative of the stochastic behavior of systems that exhibit sudden and large fluctuations. Such fluctuations arise from super-diffusive processes, such as Levy flights, where rare events can occur. The frequency of these rare events is controlled by α, which governs the spatiotemporal correlations within the systems. We see, therefore, that α not only tunes the range of avalanches and sub-avalanches but also controls the spatiotemporal correlations. In our model, the active motions induce abrupt increases in the size of avalanche clusters, resulting in discontinuous jumps in avalanche growth. To clarify, consider a sub-avalanche of size $s_{1}$ located near another sub-avalanche of size $s_{2}$, only separated by one unit of distance. If these two groups move actively towards each other, causing an avalanche at that site, they become connected. As a result, the size of the combined group at the next time step (T+1) is $s_{1}+s_{2}+1$. This means that all geometric properties of the system experience a “flight” at that time. We hypothesize that these flights are responsible for the occurrence of fat-tailed distributions in our model, indicating that active motions not only influence the range of avalanches and sub-avalanches but also control the spatiotemporal correlations within the system.

Although our ant pile model does not exhibit a fractal structure for a general ζ, it is still essential to investigate how this structure is retrieved when we approach the limit ζ→0. An important relation that holds for fractal objects is the *l*-$r_{g}$ relation. For a fractal object, one expects that [55]
(11)$$\langle \log l\rangle =D_{f}\langle \log r_{g}\rangle +\mathrm{cont}.,$$
where $D_{f}$ is the fractal dimension and is the ensemble average. For the BTW model, we know that $D_{f}^{\mathrm{BTW}}=\frac{5}{4}$ [56]. In order to examine how this relation changes in our ant pile model, we utilize the equivalence of the probability measures expressed as $p\left(l\right)dl=p\left(r_{g}\right)dr_{g}$. By combining this with Equation (Equation 3), the following equation can be derived:(12)$$\frac{dl}{dr_{g}}\propto \exp\left[{\left(\lambda_{l}l\right)}^{\alpha_{l}}-{\left(\lambda_{r_{g}^{2}}r_{g}^{2}\right)}^{\alpha_{r_{g}^{2}}}\right].$$For ζ→0, this relation is a power law, i.e., the distribution functions behave similarly to p(y), leading to a hyper-scaling relation among $\tau_{l}$, $\tau_{r_{g}}$, and $D_{f}^{\mathrm{BTW}}$ [55]:(13)$$D_{f}^{\mathrm{BTW}}=\frac{\tau_{r_{g}}-1}{\tau_{l}-1}$$
where $\tau_{r_{g}}$ and $\tau_{l}$, given in Equation (Equation 5), are the exponents corresponding to the gyration radius and loop length in the BTW limit (note that $\tau_{r_{g}}$ is obtained using Equation (Equation 5) for $\alpha_{r_{g}}\equiv 2\alpha_{r_{g}^{2}}$). Note that the exponents of this equation are not well-defined for a general ζ value, since $\tau_{l}$, $\tau_{r_{g}}$, and $D_{f}$ are only definable in the case where ζ→0, where power-law scaling relations are restored, i.e., only in the BTW limit, where this relation is famous to be valid. Figure 6 presents the simulation results for $\langle \log l\rangle$ in terms of $\langle \log r_{g}\rangle$ and the analytical solution to Equation (Equation 12). For small amounts of ζ, and small length scales ($r_{g}$), it turns out to obey the power-law behavior in Equation (Equation 11). It is observed that the fractal dimension, represented by the main figure’s slope, approaches the expected value for the BTW model in the limit ζ→0. Interestingly, as the value of ζ increases, the range of linear (fractal) behavior decreases, indicating that the avalanches become more “self-similar“ and “fractal” as ζ decreases. Moreover, after undergoing a jump, $D_{f}$ decreases with the increase in ζ. This suggests that the external boundaries of the avalanches become less “twisted” or “rough” for larger ζ values.

## 3. Conclusions

To conclude, we introduced an active pile model, which self-organizes itself in an anomalous stretched exponential distribution (SED) state. Parameter ζ controls the strength of “activity” in the system, so that ζ→0 retrieves the ordinary BTW model. The mass and the size of avalanches exhibit Gaussian distribution functions, with fluctuations diverging as ζ→0. This shows that the critical behaviors (SOC state) are retrieved in this limit, which is expected for the BTW model. By analyzing the sub-avalanches (disconnected components of avalanches), we identified a crossover point, $\zeta^{*}$, where the system transitions from BTW-like behavior ($\zeta <\zeta^{*}$) to SED behavior ($\zeta >\zeta^{*}$). Using the stretching exponents, we established a connection to α-stable Levy systems. The mechanism responsible for this is proposed to be abrupt jumps of the growing avalanche clusters, which increase with the increase in ζ. Using the analytical *l*-*r* relation and simulation results, we analyzed the loop fractal dimension and consistently found the expected behavior as ζ→0.

## Figures and Tables

**Figure 1 entropy-25-00861-f001:**
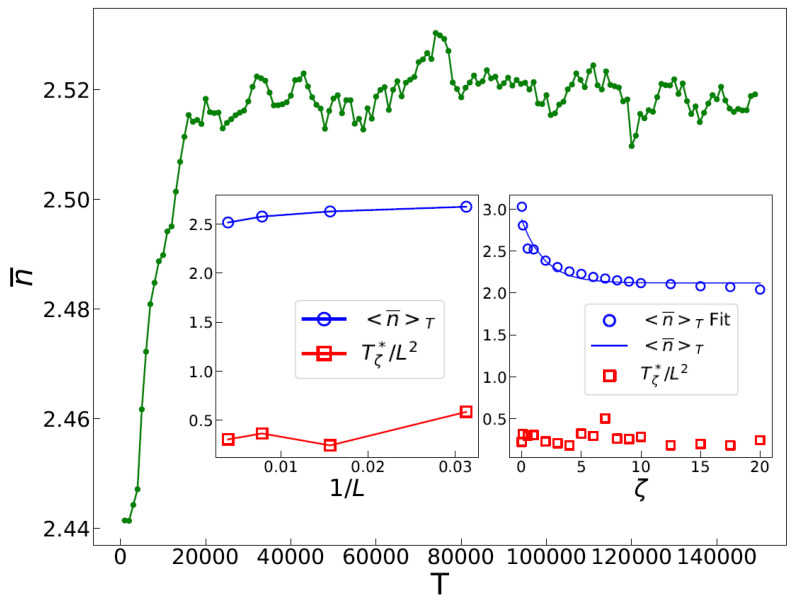
The time evolution of $\bar{n}$ versus time *T*. The left inset shows $T_{\zeta }^{*}/L^{2}$ and ${\langle \bar{n}\rangle }_{T}$ in terms of 1/L, with ζ=1. The small change in $T_{\zeta }^{*}/L^{2}$ versus *L* indicates that $T_{\zeta }^{*}\propto L^{2}$, similar to the ordinary BTW model [53]. The right inset shows $T_{\zeta }^{*}/L^{2}$ and ${\langle \bar{n}\rangle }_{T}$ in terms of ζ, with L=256. The fitting for $\langle \bar{n}\rangle$ is according to Equation (Equation 1), with $\gamma_{{\langle \bar{n}\rangle }_{T}}=0.53\pm 0.02$. Although $T_{\zeta }^{*}/L^{2}$ exhibits fluctuations, they can be attributed to the statistical uncertainty in its determination.

**Figure 2 entropy-25-00861-f002:**
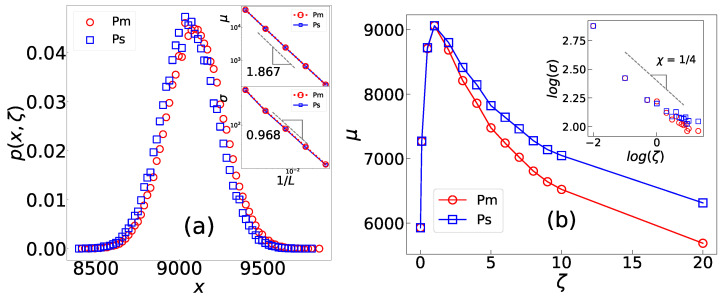
(**a**) The distribution function of the avalanche size (p(s,ζ)) and mass (p(m,ζ)) with parameters μ and σ defined in Equation (Equation 2) in terms of 1/L (insets), with ζ=1 and L=256. The insets demonstrate that μ and σ show power-law behavior in terms of system size *L*. (**b**) $\mu_{m}$, $\mu_{s}$, $\sigma_{m}$, and $\sigma_{s}$ in terms of ζ, with L=256.

**Figure 3 entropy-25-00861-f003:**
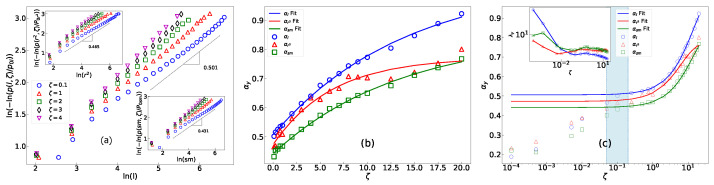
(**a**) $\ln(-\ln\frac{p(y,\zeta )}{p_{0y}})$ in terms of lny, where y=l (main), $y=r_{g}^{2}$ (upper inset), and y=sm (lower inset), with L=256. (**b**) $\alpha_{y}$ in terms of ζ, with the corresponding fits for L=256. (**c**) $\alpha_{y}$ and $\lambda_{y}$ (inset) in terms of ζ on semilog scale, with L=256. The light blue box shows the crossover region to the BTW fixed point.

**Figure 4 entropy-25-00861-f004:**
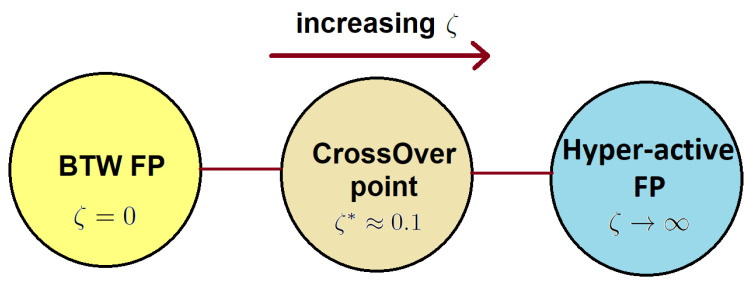
Phase diagram of the ant pile model. FP shows a fixed point. There are two FPs: the BTW FP (ζ→0) and the hyper-active (ζ→∞) FP; there is a crossover point between them around $\zeta^{*}\approx 0.1$.

**Figure 5 entropy-25-00861-f005:**
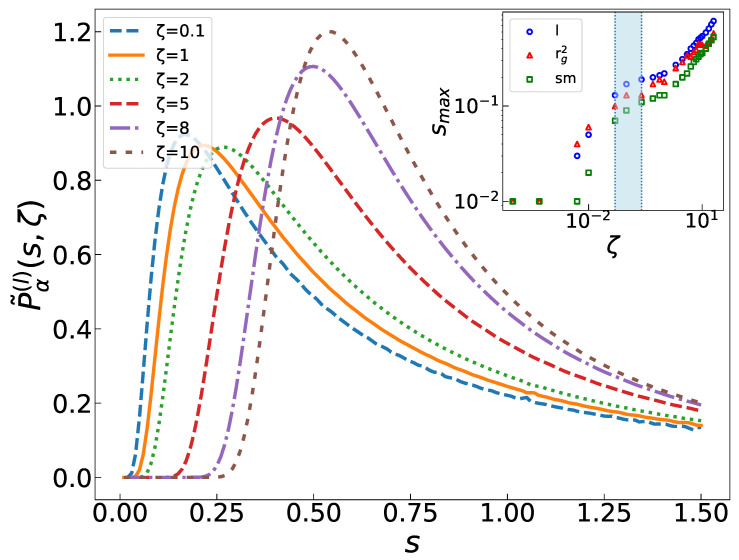
${\tilde{p}}_{\alpha }^{\left(l\right)}\left(s,\zeta \right)$ in terms of *s* for *l*. Inset shows peak point $s_{\max}$ in terms of ζ for all three cases. A crossover is observed around $\zeta^{*}\approx 0.1$ (the blue region).

**Figure 6 entropy-25-00861-f006:**
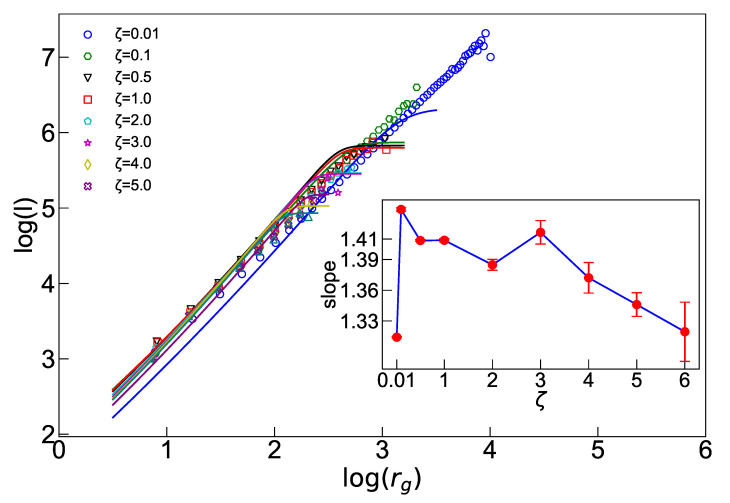
The fractal structure of avalanches. The main figure is $\langle \log l\rangle$ in terms of $\langle \log r_{g}\rangle$, with slope $D_{f}$ for L=256. Inset shows $D_{f}$ in terms of ζ.

## Data Availability

The research data supporting this publication are provided within this paper.

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
