# Peer review of "Anomalous Self-Organization in Active Piles"

_entropy, 2023, doi:10.3390/e25060861_

Round 1

Reviewer 1 Report

Report on the paper "Anomalous Self-Organization in Active Piles"
by M. Nattagh-Najafi, M. Nabil, R. H. Mridha and S. A. Nabavizadeh.

Summary
In this paper, the authors introduce and study the behavior of a model of active piles, the ant-piles, generalizing the classical BTW sandpile model. In the proposed model, a fraction of the particles can independently diffuse through the lattice, in between two consecutive addition of particles. An activity parameter fixes the number of diffusive movements allowed between successive particle additions. The authors characterize the system's behavior by means of the distribution of mass and size of the avalanches and sub-avalanches, as well as some geometric features of the sub-avalanches. Interestingly, the distributions of global characteristics obey a normal law, with parameters varying in a nontrivial way with the activity. In particular, the variance following a power-law behavior for high values of the activity parameter.  Yet more interesting is the fact that the local characteristics, associated with sub-avalanches follow stretched exponential distributions, with exponent depending monotonously with the activity parameter, but suffering a noticeable transition for small values of the parameter, consistent with the retrieving of the classical SOC dynamics at zero activity.

General comment
In my opinion, the work is relevant and will certainly attract the interest of the experts. It concentrates on numerical results and further analytical studies will certainly follow, but it has already pointed out several interesting lines for further research.

Recommendation
I recommend its acceptance after some clarifications are made and some minor mistakes are corrected.

Corrections and suggestions
The phrase "Self-organized criticality can be found ... " in lines 26 and 27, seems to be misplaced. It should be erased.

In line 68, it should say "... four ants that ARE either distributed to the neighboring sites .."

In the caption of Figure 1, "On the right inset shows..." should be replaced by "The right inset shows... "

In line 150, "In the left inset of Fig. 1, presents..." should be replaced by "The left inset of Fig. 1 presents..."

In line 194, it is mentioned in parenthesis that the sub-avalanches are the disconnected components of the avalanches. It is rather the opposite. The connected components of an avalanche, which is a global event occupying, in general, a disconnected region of the lattice, are the sub-avalanches. This should be corrected.

When introducing the geometric characteristics of a sub-avalanche, such as the "loop length" (the perimeter), and the gyration radius, it would be very useful to properly define them. In particular, the gyration radius requires a small explanation.

In line 233, after "dynamics and complexity." The next word should start with a capital.

In line 305, although it is obvious from the context,  $\alpha_r$ should be replaced by $\alpha_{r_g}$. Otherwise, the notation $r$ for the gyration radius can be adopted from the beginning.

In Equation (8), the $z$ in the first exponent has to be replaced by $s$.

In line 334, in the part "...is controlled by $\alpha$, governs...", it seems to be something missing. Maybe "...is controlled by $\alpha$, and it governs..."

Prior to line 370 and in Equation (11), the nature of the exponents $\tau_l$ and $\tau_r$ should be explained.

The relationship between the Levy distribution and the stretched exponential distribution is known. I found a relatively recent reference: "Górska, et al. The stretched exponential behavior and its underlying dynamics. The phenomenological approach. Fractional Calculus and Applied Analysis, 20(1), (2017)". It is mentioned in this reference that the cited relation can be traced back to " H. Pollard, The representation of $e^{−x\lambda}$ as a Laplace integral. Bull. Amer. Math. Soc. 52 (1946), 908–910.  I suppose there are more pertinent references concerning this relationship, that could be cited in the paper.  

Author Response

We would like to thank the referee for the constructive comments, and the detailed analysis of the manuscript, which encouraged us to revise our manuscript. We addressed the comments of the referee point by point accordingly. The changes are highlighted in blue in the manuscript.

First of all, we should thank the referee for stating that “I recommend its acceptance after some clarifications are made and some minor mistakes are corrected.”

  • The phrase "Self-organized criticality can be found ... " in lines 26 and 27, seems to be misplaced. It should be erased. In line 68, it should say "... four ants that ARE either distributed to the neighboring sites ..". In the caption of Figure 1, "On the right inset shows..." should be replaced by "The right inset shows... ". In line 150, "In the left inset of Fig. 1, presents..." should be replaced by "The left inset of Fig. 1 presents...". In line 194, it is mentioned in parenthesis that the sub-avalanches are the disconnected components of the avalanches. It is rather the opposite. The connected components of an avalanche, which is a global event occupying, in general, a disconnected region of the lattice, are the sub-avalanches. This should be corrected. In line 233, after "dynamics and complexity." The next word should start with a capital. In line 305, although it is obvious from the context, $\alpha_r$ should be replaced by $\alpha_{r_g}$. Otherwise, the notation $r$ for the gyration radius can be adopted from the beginning. In Equation (8), the $z$ in the first exponent has to be replaced by $s$. In line 334, in the part "...is controlled by $\alpha$, governs...", it seems to be something missing. Maybe "...is controlled by $\alpha$, and it governs...".

We would like to thank the referee for focusing in the text, and letting us about these issues. We resolved all the mentioned problems with the manuscript in the new version.

  • When introducing the geometric characteristics of a sub-avalanche, such as the "loop length" (the perimeter), and the gyration radius, it would be very useful to properly define them. In particular, the gyration radius requires a small explanation.

We added a new description on page 3 to define the geometrical observables.

  • Prior to line 370 and in Equation (11), the nature of the exponents $\tau_l$ and $\tau_r$ should be explained.

It is an important point. To resolve it, we added equations 4 and 5 to the new version. Now the definitions are clear. We added a sentence after Eq. 13.

  • The relationship between the Levy distribution and the stretched exponential distribution is known. I found a relatively recent reference: "Górska, et al. The stretched exponential behavior and its underlying dynamics. The phenomenological approach. Fractional Calculus and Applied Analysis, 20(1), (2017)". It is mentioned in this reference that the cited relation can be traced back to " H. Pollard, The representation of $e^{−x\lambda}$ as a Laplace integral. Bull. Amer. Math. Soc. 52 (1946), 908–910. I suppose there are more pertinent references concerning this relationship, that could be cited in the paper.

We understand the referee’s concern and add the proposed references. We also add a new reference that concentrates on this connection.

Reviewer 2 Report

Referee report on the paper

"Anomalous Self-Organization in Active Piles"

by Morteza Nattagh-Najafi et al.

The Authors propose and sudy a model that is supposed to describe a kind of "active" self-organized criticality (SOC). This phenomenon attracted much attentin in recent years, so the topic of the paper is actual. The model comprises random walk of particles on a two-dimensional lattice and threshold topping. The model involves a tuning parameter \zeta, interpreted  as "degree of activity" and ranging from very small values to 20. 

The Authors performed detailed numerical investigation of the model. The main finging is the following. The system evolves to a kind of self-organized critical state. For small \zeta it belongs to the universality class of the well-known Bak-Tang-Wiesenfeld (BTW) model. When \zeta increases, a phase transition to the new SOC state takes place at a certain critical value \zeta_c approximately equal to 0.1.

The probability density function for this new "active" state is described by a "stretched exponential with a fat tail". Using the Laplace transformation, the Authors decompose it into degrees of freedom with exponential decay with well-defined attenuation coefficients. Using these stretching exponents, the Authors established a connection with the alpha-stable Levy systems.

I believe that these results are interesting and the paper can be published in "Entropy". 

I have some small remarks.

1) The abbreviation BTW in the abstract and the text should be explained.

2) Interpretation in terms of ant population seems rather far-fetched to me. 

However, once it is accepted: 

in line 87  "the number of sand grains" should be "the number of ants"?

3) line 174: the apostrophe probably means \zeta's ?

Author Response

We would like to thank the referee for the constructive comments, and the detailed analysis of the manuscript, which encouraged us to revise our manuscript. We addressed the comments of the referee point by point accordingly. Our changes in the text, and our response here in this document are highlighted in blue.

First of all, we should thank the referee for stating that “I believe that these results are interesting and the paper can be published in "Entropy".”

  • The abbreviation BTW in the abstract and the text should be explained.

We explained it in the new version.

  • Interpretation in terms of ant population seems rather far-fetched to me. However, once it is accepted: in line 87 "the number of sand grains" should be "the number of ants"?

We thank the referee for letting us know about this. We revise thoroughly this part, and hope that it is now clearer.

  • line 174: the apostrophe probably means \zeta's ?

Yes, we correct it.

Author Response

We would like to thank the referee for the constructive comments, and the detailed analysis of the manuscript, which encouraged us to revise our manuscript. We addressed the comments of the referee point by point accordingly. We highlighted (in blue) the changes in the main text.

First of all, we should thank the referee for stating that “The results seem to be original and well presented, and I have only minor comments.”

  • Although I understand that in the introduction are given the usual examples of presumed SOC, the present understanding in the field is that, in contrast to conservative systems as sandpiles and extremal models, dissipative models of earthquakes, forest fires and neuronal avalanches are not true SOC, but systems that present approximate SOC (Self-organized quasicriticality or SOqC, or also Homeostatic Criticality). So, perhaps the authors could cite this more modern literature – search for SOC and Homeostatic criticality in the Google Scholar).

We understand the referee’s concern and thank them for this constructive comment. We revised this part in the new version of the manuscript.

  • In Fig. 5 we have not the important curve for ? = 0.1.

The referee is completely right. We add this plot to figure 5 in the new version.

  • Line 371: It seems to me that the plot of Fig. 6 refers to Eq. 9, not Eq. 10.

In Fig. 6 we are presenting the simulation results for <log l> in terms of <log r_g>, and for the analytic fitting, we use the Eq. 10. For small \zeta values, it turns out to obey the power-law relation Eq. 9. We corrected this in the paper.

  • It could be interesting to show that Eq. 11 holds with the actual values of the measured exponents.

It is a good suggestion. However, the exponents of this equation are not well-defined for a general \zeta value, since \tau and D_f are only definable for the case \zetaà0 where power-law scaling relations are restored, i.e. only in the BTW limit, where this relation is famous to be valid. We add some comments about it in the new version.

  • Why in the inset of Fig. 6 we have not the data for the ? = 0.1 case? It is closer to 5/4 or not?

We have added \zeta=0.1 to the new version of this Figure. Yes, it is closer to 5/4. In the old figure, there was a shift in the horizontal axis, that we corrected in the new version.

  • Line 25: rice instead of rise.

We correct it.

  • 1, 2, 3, 5: The inset legends are somewhat very small.

We resize them ad referee suggested,

  • Line 166: ?2 " instead of ?.

Thanks for catching the typo. We correct it.

  • 3c is a plot linear-log, not log-log.

We correct it.

  • A revision must be done in the formatting of the references. Problems with the journal’s names: References 1, 2, 3, 4, 5, 9, 11, 32, 39, 42, 45,47.

We revised the references.

  • Two old Arxiv papers, to see if they have been published: 25, 33.

Reference 33 (40 in the new version) is updated, but for the Ref. [25] (Ref. [25] in the new version), no journal was found.
